# Nutrition and Supplementation in Ulcerative Colitis

**DOI:** 10.3390/nu14122469

**Published:** 2022-06-14

**Authors:** Marcelina Radziszewska, Joanna Smarkusz-Zarzecka, Lucyna Ostrowska, Damian Pogodziński

**Affiliations:** Department of Dietetics and Clinical Nutrition, Medical University of Bialystok, Ul. Mieszka I 4B, 15-054 Bialystok, Poland; joanna.smarkusz-zarzecka@umb.edu.pl (J.S.-Z.); lucyna.ostrowska@umb.edu.pl (L.O.); damian.pogodzinski@umb.edu.pl (D.P.)

**Keywords:** ulcerative colitis, colitis ulcerosa, diet, Mediterranean diet, UCED, SCD, FODMAP, supplements

## Abstract

Ulcerative colitis (UC) belongs to the group of inflammatory bowel diseases (IBD). UC is an incurable, diffuse, and chronic inflammatory process of the colonic mucosa with alternating periods of exacerbation and remission. This review aimed to analyze the scientific research conducted to date to determine what impact different nutritional plans and dietary supplements may have on the course of UC. The latest 98 articles about nutrition and supplementation in ulcerative colitis were used to prepare the work. Certain components in food can greatly influence the course of UC, inducing changes in the composition and function of the gut microbiome. This activity may be an important part of therapy for people with IBD. The Mediterranean diet has shown the most promising results in the treatment of patients with UC due to its high content of biologically active foods. Patients with UC may benefit from the UC Exclusion Diet (UCED); however, it is a new nutritional plan that requires further research. Patents frequently resort to unconventional diets, which, because of their frequent elimination of nutrient-rich foods, can worsen the health and nutritional status of those who follow them. The benefits of omega-3 fatty acids and probiotics supplementation may have additional therapeutic effects; however, the evidence is not unequivocal.

## 1. Introduction

Ulcerative colitis (UC) belongs to the group of inflammatory bowel diseases (IBD) [1]. UC is a, incurable, diffuse, and chronic inflammatory process of the colonic mucosa with alternating periods of exacerbation and remission [2,3]. UC has been recognized as a global disease because the incidence is steadily increasing worldwide [2], with the highest reported in Northern Europe, Canada, and Australia [3]. UC is most commonly diagnosed between the ages of 30 and 40 [3]. The exact etiology of UC is not known. However, it is possible that genetic, immunological, and environmental factors contribute to the disease pathogenesis [4,5].

The most common presenting symptoms of UC are bloody stools [4]. Associated symptoms may also include urgency to defecate, abundant rectal mucus excretion, increased frequency of bowel movements, nocturnal bowel movements, abdominal discomfort (pain, cramping), urinary incontinence, fatigue, fever, dehydration, and malnutrition [3,4,6]. Tenderness, soreness, and abdominal bloating may be noted on palpation, as well as blood on rectal examination [3]. The number of stools passed, as well as the presence of other symptoms can vary. Symptoms may reflect the severity of intestinal mucosal inflammation and the extent of changes [3,4]. In addition, the results of the tests performed may show signs of anemia [4]. Axial or peripheral arthropathy, scleritis, and erythema nodosum appear to be the most common extraintestinal manifestations [4]. Moreover, primary sclerosing cholangitis (PSC) is a rare but poor prognostic complication of UC. Patients also have an increased risk of developing colorectal cancer [6].

According to the current state of knowledge, there is no gold standard for the diagnosis of UC [6]. The current marker used to rule out or confirm IBD is fecal calprotectin levels. Low levels of this indicator indicate a less than 1% probability of developing IBD [3]. The final diagnosis is based on clinical presentation, biochemical and histological findings, as well as endoscopic findings, which include colonoscopy and rectoscopy for patients with a severe flare of disease [4,6].

The form of treatment for UC is tailored to the severity, distribution, and type of disease, including its course, patient’s response to prior medications and side effects, relapse rate, and extraintestinal symptoms. The age of the patient at diagnosis and the duration of the disease are also significant variables [7]. The main goal of treatment is to obtain clinical remission confirmed by endoscopic examination, without the need to start treatment with steroids [6]. The treatment of choice in most cases is mesalazine, to which aminosalicylate enema is added in some patients [6,7]. In the absence of response, systemic corticosteroids are included in the therapy, which are also recommended for patients with severe UC flare-ups [6,7]. Biologic or immunosuppressive drugs present an alternative treatment option for patients with UC when previous pharmaceuticals have not worked [7]. Surgery is the ultimate treatment for UC, used when pharmacological treatment fails and when serious complications of the disease occur [6].

Due to the specificity of UC, besides pharmacological and surgical treatment, introduction of appropriate dietary and nutritional habits is an extremely important element of the therapy, which, however, is still underestimated and often omitted in medical practice [1,8]. Despite the lack of specific dietary advice in IBD, even more than 70% of sufferers note that inadequate nutrition significantly affects the course of the disease and increases the frequency and severity of symptoms [8,9]. Consequently, patients with UC intensively seek nutritional guidance to help improve their quality of life and contribute to symptom relief [10]. Unfortunately, studies to date do not provide a solid basis for creating strong evidence-based dietary recommendations [8,10]. Patients’ curiosity about diet and lack of precise recommendations compel them to seek information from the Internet and other non-medical sources [10,11]. The purpose of this study was to review research published to date on the effects of different dietary models on the course of UC.

## 2. Method 

A systematic search of the literature was conducted in PubMed base to identify studies relevant to the current review. The following search string was applied: (“ulcerative colitis” OR “colitis ulcerosa” OR “IBD” OR “microbiome” OR “bowel diseases” OR “gut cancer”) and (“treatment” OR “diet” OR “Mediterranean diet” OR “fiber” OR “vegetables” OR “fruits” OR “legumes” OR “cereal products” OR “lipids” OR “fats” OR “fish” OR “omega-3 fatty acids” OR “meat” OR “protein” OR “processed meat” OR “processed foods” OR “dairy products” OR “fermented dairy products” OR “spices” OR “herbs” OR “remission” OR “exacerbation” OR “UCED” OR “SCD” OR “FODMAP” OR “supplementation” OR “iron” OR “vitamin D” OR “calcium” OR “vitamin B_9_” OR “probiotics”). We tried to limit the search for papers in the 5-year range; however, if no data were available in this range, studies from earlier years were also included. From the 1500 items found, 10 Practise Guideline, 46 Review, and 42 Clinical Trial were selected for the preparation of the paper.

## 3. The Mediterranean Diet

One of the safer dietary models, additionally recommended by ESPEN (*European Society for Clinical Nutrition and Metabolism*) as an optional method for UC patients, is the Mediterranean Diet [12]. This diet based on a high intake of vegetables and fruits as well as legumes and whole grains, which are rich in antioxidants and dietary fiber as well as nuts, fish, and olive oil abundant in monounsaturated and polyunsaturated fatty acids [13]. The Mediterranean diet also includes moderate amounts of dairy, especially fermented dairy products (such as yogurt, kefir, cheese) and eggs [14]. In exchange for red meat, whose consumption should be reduced, this diet endorses the consumption of its leaner counterparts, such as turkey, chicken, and rabbit [14]. Each dish is further enriched with herbs and spices (including parsley, thyme, oregano, basil, cumin, cinnamon, turmeric), whose antioxidant effects have been confirmed scientifically [14]. 

Diet is extremely important for every human being, especially those dealing with IBD, as it is, among other things, one of the primary factors regulating the gut microbiota [1]. The modern Western diet, which is high in saturated fats and simple sugars and low in dietary fiber, may lead to a disparity in the composition of the gut microbiota [15]. Dysbiosis can cause many health disorders because the microorganisms found in the gut play an important role in the functioning of the body’s immune system [15]. Thus, the development of unfavorable microorganisms, which may be stimulated by an inadequate diet, results in the impairment of immune system function and thus may lead to increased production of inflammatory factors, which in turn stimulates the development of intestinal inflammation (leading to an increase in serum C-reactive protein and fecal calprotectin levels) [1,15]. These disorders can be modified by incorporating dietary changes. Adherence to the Mediterranean diet, as shown by studies, improves the ratio between pathogenic microorganisms such as *Firmicutes* and *Eschericha coli*, and beneficial bacteria including *Bifidobacterium* and *Bacteroides fragilis* [16,17]. It is worth noting that an increase in intestinal colonization by unfavorable microorganisms, such as those of the genera *Fusobacterium*, *Peptostreptococcus*, *Bacteroides vulgatus*, *and Bacteroides thetaiotaomicron,* and a decrease in probiotic bacteria, such as *Lachnospiraceae*, *Bifidobacterium animalis*, and *Streptococcus thermophilus*, may promote inflammation, DNA damage, and cancer cell proliferation, which contribute to the development of colorectal cancer [18]. Due to the beneficial effects of the products consumed in the Mediterranean diet on the composition of the gut microbiota, this nutritional plan may be a helpful component of therapeutic management in patients with UC [13]. 

Studies show that the use of the Mediterranean diet in UC patients after reconstructive proctocolectomy with ileo-rectal anastomosis with a created intestinal reservoir and in children with IBD, respectively, has a positive effect on fecal calprotectin levels [19,20]. A recent prospective interventional study published in 2021 evaluated how a 6-month Mediterranean diet applied in patients with IBD affects their nutritional status, quality of life, disease activity, and steatohepatitis, which often accompanies IBD [1]. The study included 142 adults with IBD (including 84 with UC). At the beginning of the study, the participants were given a dietary consultation, during which they received advice on how to change their current eating habits in order to eat according to the Mediterranean diet model. Researchers have noticed that the diet in patients with UC has many beneficial effects [1]. There were significant improvements in quality of life and indicators of malnutrition in the participants. The number of patients with active inflammation decreased after 6 months on the diet from 24% to 7%, the number of patients with high C-reactive protein levels was reduced from 50% to 37.5%, and the number of patients with fecal calprotectin levels greater than 250 mg/kg was reduced from 44% to 28%. In addition, it was noted that patients had improved anthropometric indices correlating with steatohepatitis and metabolic syndrome [1]. Inferring from the cited studies, the Mediterranean diet requires further study to increase the knowledge of its effects on IBD, but it seems to be a future way of eating for patients, bringing valuable therapeutic effects.

According to the Mediterranean diet scheme, the main foods included in the diet should be vegetables, followed by fruits. These products are a crucial source of vitamins and minerals [13]. Of the vegetables, patients with UC should mainly choose those that are not rich in water-insoluble dietary fiber, including beets, potatoes, carrots, and zucchini, which can be eaten cooked and, if well tolerated, raw. However, others rich in dietary fiber of the water-insoluble fraction, such as brassica vegetables, broccoli, and peppers, should be boiled and pureed or blended. The same should be done when incorporating fruits into the menu. Initially, those not rich in water-insoluble fiber fractions, such as bananas or apples, are recommended. If raw fruits, low in dietary fiber of the water-insoluble fraction, are well tolerated by the patient (no gastrointestinal symptoms) it is possible to introduce the consumption of other seasonal fruits. However, they should be incorporated carefully and gradually depending on the individual tolerance of UC patients [13]. A very good alternative, especially in the period of disease exacerbation, is the consumption of freshly squeezed vegetable or fruit juices, which contain all valuable nutrients contained in the product (vitamins, minerals), but lack dietary fiber insoluble in water [13]. When choosing plant foods, it is also worth paying attention to legumes, which are also a valuable source of vitamins, minerals, and dietary fiber.

Legumes are a very good source of plant protein with high biological value and complex carbohydrates, while being low in fat [21]. These products, when properly composed with cereal products to complement the amino acid composition of legume seed proteins, can be an alternative choice to high meat consumption, especially red meat [21]. However, it is recommended to consume them peeled or after squeezing them through a sieve, preceded by long cooking [13]. It is also very important to take into account other technological and thermal procedures in the preparation of dishes based on legumes, which may deprive them of fiber of the water-insoluble fraction while retaining the water-soluble fraction, causing a reduction in the content of antinutrients and thus improving their nutritional value and digestibility [22]. A review by Satya et al. noted that sprouting is one of the best legume seed treatments to induce the previously mentioned changes [22]. Comparable effects were noted for thermal treatments, which include treatments such as plain or pressure cooking and microwave preparation. The review also evaluated the benefits of prior soaking of legumes and concluded that soaking the seeds in water or 0.1% citric acid solution for 9 h is appropriate for the best preservation of vitamins and other nutrients [22]. Among the legumes that patients with UC should especially pay attention to are peas and red lentils, which have the best digestibility [13]. Lentils are rich in complex carbohydrates, including mainly starch, and are characterized among other legume seeds by the highest protein, high iron, zinc, and calcium content, while having the lowest fat content [23]. Additionally, very important is the low content of anti-nutritional substances (e.g., protease inhibitors) in this type of legume seed. Due to its soft seed coating, it requires less cooking time than other legumes [23]. Therefore, it seems reasonable to recommend that patients with UC eat legumes, especially red lentils. However, their inclusion in the menu should take into account the period of exacerbation of the disease, symptoms, and individual tolerance. Particular attention should be paid to adequate prior technological and thermal treatment, as well as the amount of food consumed. It is best to start with smaller portions. When choosing vegetables, fruits, and legumes, the dietary fiber content should be taken into account.

Dietary fiber is an important component of the diet, but an important issue is which fraction of fiber is present in the product (water-soluble or water-insoluble fraction) because, depending on the type, it can adversely affect the intestinal mucosa and is therefore not always recommended for patients with UC. 

All elements of dietary fiber of the water-soluble fraction, including inulin, pectin, gums, and beta-glucans, do not irritate the colonic mucosa, while stimulating the growth of microorganisms producing butyric acid and propionic acid in the lumen of the large intestine. These substances show protective activity against the intestinal mucosa, and butyric acid additionally inhibits the synthesis of pro-inflammatory factors [13,14,24,25]. A study conducted by Desai et al. in mouse models demonstrated the significant effect of dietary fiber on the occurrence and treatment of intestinal pathology [26]. The basic protective element of the intestines is the mucous membrane and the microorganisms living within it, whose condition and function depend on the intake of dietary fiber. Deprivation of access to dietary fiber for the microorganisms colonizing the intestines, which occurs with a low-fiber diet, leads to energy extraction from the breakdown of components contained in the mucus secreted by the human body, resulting in a loss of integrity of the colonic mucosa [26]. This leads to the penetration of unfavorable microorganisms across the intestinal barrier, inducing inflammation within the gut, including UC [24,26].

The type of fiber is especially important to consider in patients with exacerbation of UC, existing diarrhea, or abdominal pain. It is recognized that an undeniable recommendation for this group of individuals is to limit dietary fiber intake [27]. However, pectins, because of their properties, may be recommended in the form of cooked, mashed vegetables. According to the World Gastroenterology Organization guidelines, a diet rich in dietary fiber should be introduced UC patients with rectal inflammation accompanied by constipation [27]. Some studies have shown that the inclusion of dietary fiber in the diet of patients in remission of UC may have beneficial effects [12]. A comparison by Fernández-Bañares et al. evaluated the efficacy of using plantain (10 g twice daily) and mesalazine (500 mg three times daily) for 12 months on the length of remission maintenance [28]. The study divided patients into three groups. The first group received plantain seeds, the second group received mesalazine, and the third group received combination therapy (plantain seeds and mesalazine). Recurrence of UC exacerbation, after 12 months of treatment, occurred in 30% of those using combination therapy, 40% of those using plantain seeds, and 35% of those in the mesalazine group. These results allowed the study authors to conclude that plantain seeds, and thus a diet rich in dietary fiber, may have a similar potency in maintaining remission in UC as mesalazine therapy [28]. A study by Hallert et al. also examined the effect of dietary fiber on the course of UC in patients in remission [29]. Researchers administered plantain husk to 29 people in remission for a period of 4 months. The results of their study showed that plantain husk may be effective in reducing gastrointestinal symptoms that occur during the remission period in UC patients [29]. A review of randomized trials by Wedlake et al. [30], similarly to previous studies, confirmed that dietary fiber may have a positive effect on the course of UC remission. However, the authors pointed out that further clinical studies are necessary to confirm the hypothesized anti-inflammatory role of this component. In contrast, ESPEN observes that the studies conducted are not sufficient to make a general statement about the safety of increased dietary fiber in the diet of patients with UC. Therefore, they believe that fiber-rich diets should not be universally recommended for this group of patients to maintain remission [12].

Vegetables, fruits, and legumes form the basis of the Mediterranean diet [14]. When properly selected and prepared, they are very important in the diet in UC patients because they are a rich source of vitamins, minerals, and dietary fiber of the water-soluble fraction [13,24]. A meta-analysis by Li et al. found that fruit and vegetable consumption was associated with a lower risk of UC [25]. This association correlates with the content of beta-carotene in these products, which influences the synthesis of biologically active compounds involved in the alignment of intestinal cell structure and positively affectsthe human body, as well as flavonoids, which are suspected to participate in maintaining the integrity of colonocytes, and dietary fiber of the soluble fraction [24,25]. It is an undeniable fact that these products are an indispensable part of the diet; however, everyone suffering from IBD should remember to diversify their diet and therefore it is important to include other food products, such as cereals.

Unfortunately, research to date provides little information regarding the consumption of cereal products, and it is not entirely clear which types of cereals may be particularly valuable for people with UC. The possibility of using groats, pasta, and bread made from standard varieties of cereals such as durum has been evaluated. The aforementioned cereals are some of the best choices because they have not been genetically modified, and thus they do not trigger an immune response and inflammation in the gastrointestinal tract [13]. Of particular note is rice, which has the potential to alleviate symptoms of intestinal mucosal inflammation [13]. Often, patients with IBD wonder about the possibility of using gluten-free grain products to reduce the severity of symptoms and risk of disease flare-ups. However, there is still a lack of studies showing an association between the consumption of gluten-containing products and the course of IBD. Therefore, gluten-free diets should not be recommended to this group of patients, because excluding these products results in elimination of the source of other nutrients, such as vitamins and minerals. Cereal products, vegetables, fruits, and legumes are the main sources of carbohydrates in the diet. However, the body also needs other nutrients to function properly, such as fats, of which olive oil should be the main source.

Cold-pressed olive oil is the primary source of fats in the Mediterranean diet [14]. This product is a rich source of phenolic compounds and oleic acid [14]. These components, by affecting the stability of the intestinal epithelium and strongly inhibiting the synthesis of oxidatively acting reactive oxygen species, play an extremely important role in preventing the development of inflammation in the intestines and cancer [24,31]. The vast majority of these compounds interact directly with the intestinal tract during the process of absorption and accompanying metabolism. In contrast, some undergo complex chemical transformations leading to the production of various metabolically active products that may exhibit even more potent biological activity than the primary compounds [31]. Reddy et al. found high levels of tert-butyl hydroperoxide in Caco-2 cells cultured in medium containing medium-chain fatty acids (MCFA). They lead to increased cell damage, decreased glutathione levels, and IL-1β stimulation of IL-8 production. In contrast to cells cultured on MCFA medium, cells grown in medium enriched with oleic acid and hydroxytyrosol, which can be found in olive oil, do not show high levels of tert-butyl hydroperoxide [32]. Researchers additionally observed that, compared with the use of a single ingredient, the combination of eicosapentaenoic acid (EPA), docosahexaenoic acid (DHA), and hydroxytyrosol reduced cell damage and IL-8 synthesis to a greater extent [32]. The study also tested the effects of these compounds in vivo in rats. Animals with induced UC were assigned to one of four groups that differed in the type of fat consumed. After 10 days of observation, the animals from the MCFA ingestion group exhibited severe active UC, and there was a consistent weight loss and diarrhea. It is worth noting that the addition of fish oil to both the MCFA-rich diet and the olive oil-dominant diet significantly attenuated ongoing intestinal inflammation, but this effect was significantly more beneficial in the olive oil group. Thus, the researchers concluded that medium-chain fatty acids, by stimulating oxidative processes and IL-8 synthesis in colonocytes, elevate susceptibility to UC. On the other hand, the use of olive oil, especially its combination with fish oil, due to the presence of oleic acid and hydroxytyrosol and omega-3 fatty acids, respectively, may have a protective and soothing effect on the intestinal mucosa in the course of UC, showing a high therapeutic potential in the course of this disease [32].

Because of the omega-3 fatty acids in fish, such as DHA and EPA, it may be beneficial to include these foods in the diet of patients with UC [13]. EPA and DHA are used to synthesize eicosanoids, such as resolvins and protectins, which exhibit anti-inflammatory effects [33,34]. Eicosanoids have the ability to mobilize neutrophils, exhibit intense chemotactic effects, and trigger proliferation of epithelial cells of the injured intestinal mucosa [34]. These acids can induce beneficial changes in the gut microbiota, and these changes consequently result in increased production of anti-inflammatory substances, such as short-chain fatty acids [35]. It is also suspected that omega-3 fatty acids in cooperation with the immune system and intestinal microbiota can probably influence the reduction of intestinal wall permeability [35]. However, there are not enough clinical studies to suggest specific amounts of omega-3 fatty acids for the inclusion of supplementation [35]. In addition to containing high-value fats, fish is also a source of complete protein, the supply of which should be ensured from various food groups. The Mediterranean diet recommends a low intake of meat in general, and red meat consumption in particular should be limited [14]. Therefore, complete protein is mainly provided by eating a variety of fermented dairy products, eggs, and lean white meat [14].

Adequate selection of protein-supplying foods may have positive effects in patients with UC, as confirmed in a study by Jowett et al. [36]. In this study, it was observed that excessive meat consumption, especially red meat and processed meat products, may contribute to the exacerbation of colonic mucosal inflammation in UC [36]. Participants in the analysis who experienced an exacerbation of UC had an average consumption of red meat and processed meat products as high as 172 g/d, and those without an exacerbation of the disease 124 g/d. Researchers believe that limiting red meat and incorporating other protein sources into the diet may prolong the duration of remission. They link the negative effects of these products on the course of UC to the high content of sulfur-containing amino acids and added sulfates in these products, respectively. Sulfur compounds in the intestine are converted to hydrogen sulfide, which causes damage to the cells of the colonic mucosa [36]. However, although other meats and fish also hold sulfur-containing amino acids, the study found that they did not have a significant effect on the intestinal mucosa. Similarly, high consumption of milk and dairy products is not associated with exacerbation of UC [36]. It was concluded that there must be other ingredients in processed meat products and red meat that can negatively affect the gut. In 2015, these products were classified by the *International Agency for Research on Cancer* (IARC) as carcinogens (Group 1) and probable carcinogens (Group 2A), respectively [37]. A review of studies conducted by WCRF (*World Cancer Research Fund International*) in conjunction with AICR (*American Institute for Cancer Research*) shows that this strong effect on cancer risk is due to the presence of iron in both red meat and processed meat products [38]. Iron in red meat can exist in a heme-bound form (this component is present in red meat at concentrations 10 times higher than in its white counterparts) and a free form released from heme [39]. Carcinogenic effects may be caused by, among other things, increased oxidative stress and damage to the intestinal mucosal epithelium, resulting in increased cytotoxicity, inflammation, and genetic mutations [39,40]. In addition, heme iron, contained in red meat, may facilitate the formation of N-nitro- compounds in the intestinal lumen, leading to DNA adducts that can cause mutations and initiate carcinogenesis [39,41]. Nitrates and nitrites added to processed meats and compounds formed by curing or smoking (e.g., polycyclic aromatic hydrocarbons) or thermal processing (especially frying and grilling) of red meat can potentiate the formation of N-nitro compounds in the body [39,41]. As a result of these observations, a group of experts concluded that each 50 g increase in processed meat consumption increases the risk of colon cancer by 16% [38].

In contrast to red meat and processed meat products, studies show that it may be beneficial to include yogurt in the daily diet of patients with UC [24]. This product is a source of beneficial microorganisms that are able to colonize the human gastrointestinal tract and thus can affect the immune system as well as the occurrence and course of UC [24]. Modifying the secretion of inflammatory factors, including stimulating the secretion of IL-10 while inhibiting the release of IL-8 in colonocytes, may indicate that some strains of bacteria contained in yogurt have anti-inflammatory effects in the course of UC [24]. Not only yogurt, but also other fermented dairy products containing lactic acid bacteria may prove to be a functional component in the prevention and treatment of IBD, as they modify the secretion of biologically active compounds by altering the function of the immune system [24]. It has been indicated that the inclusion of two servings (125 g/d) of yogurt or other fermented dairy products in the diet of patients with UC may have positive effects [24]. 

In addition to fermented dairy products, the diet of patients with UC should also include a moderate intake of milk and other forms of dairy (such as certain types of cheese), which are also part of the Mediterranean diet [14,24]. All dairy products may have an effect on the functioning of the intestinal microbiota and the structure of the intestinal mucosa and the ongoing inflammation in the course of UC [24]. The presence of saturated fatty acids in dairy products may indicate a pro-inflammatory effect; however, the protein content (including whey protein) and relevant amino acids and minerals, such as magnesium, which have anti-inflammatory properties, may alleviate the negative effects [24,42]. It is undeniable that dairy is a major source of calcium and essential nutrients, including complete protein [24,42]. Additionally, a review of meta-analyses showed that milk consumption is associated with more beneficial than adverse effects to the body [43]. Therefore, people with UC should not give up dairy products unless they have lactose intolerance or food allergies related to dairy consumption.

Not only dairy, but also eggs should be included in the meals of patients with UC [24]. According to the meta-analysis of the studies performed to date, bioactive compounds contained in them, such as phosphotidylcholine, ovotransferrin, or lysozyme, may inhibit the synthesis of inflammatory cytokines while stimulating the production of anti-inflammatory substances [24]. A study by Stremmel et al. examined whether the use of phosphatidylcholine could facilitate withdrawal of steroid therapy [44]. The researchers divided 60 people with active UC who were refractory to steroid therapy into two groups. In the study group, which used phosphatidylcholine four times/day in a total amount of 2 g for a period of 12 weeks, a higher number of remissions and improvement in disease activity indices were observed and up to 80% of subjects discontinued steroid therapy without a relapse (in the control group, only 10% of subjects) [44]. In contrast, a study by Kobayashi et al. examined how the use of ovotransferrin might affect the inflammation in the colon [45]. Researchers used supplementation of this ingredient in mice with induced IBD. Ovotransferrin was administered at 50 or 250 mg/kg b.w./d for 2 weeks. This supplementation resulted in a significant reduction of clinical symptoms, weight loss, and inflammatory cytokines and histological activity [45]. Due to the positive effects of these components on ongoing IBD, it is worth including eggs in the diet of patients with UC.

In conclusion, it seems reasonable to provide dietary recommendations to people with UC to include milk and dairy products, especially yogurt and other fermented products, and eggs in their diet because of possible beneficial effects. Moreover, the recommendation to eliminate red meat and processed meat products from the diet and use white meat and fish instead has solid grounds. In addition to a variation of basic foods that provide nutrients, each dish should be enriched, if possible, with spices and herbs that add flavor to the food but also complement it with other valuable ingredients.

Among the spices and herbs, turmeric deserves special attention. Its main active ingredient curcumin exhibits broad anti-inflammatory, antioxidant, antifungal, and antimicrobial activities, may enhance cell apoptosis, and has anticarcinogenic properties [46]. The anti-inflammatory activity of curcumin, including the support of IBD therapy, is based on the inhibition of effective myeloperoxidase function, prevention of NF-kappaB kinase and IKB activation, as well as the reduction of interleukin-1 production and neutrophil infiltration [46]. Studies have shown that combining pharmacological treatment for patients with UC with curcumin supplementation can improve disease activity indices, reduce the risk of recurrent exacerbation, and alleviate disease symptoms, chief among which are improved mood and minimized urgency of bowel movements [47,48,49,50,51]. Despite the promising results of curcumin supplementation in supporting the treatment of UC, researchers note that there is a need for more studies to confirm the efficacy and safety of its higher doses, but a small addition of curcumin can add variety to daily meals [46,47,48].

The Mediterranean diet, due to its specific composition, may therefore be an appropriate dietary choice for people with UC. However, it is necessary to consider individual modifications of this diet, by choosing appropriate products, as well as techniques of technological and thermal processing of the selected assortment. A well-composed diet will provide the appropriate energy as well as vitamins and minerals and other biologically active compounds, the intake of which may be of particular benefit to the health of people struggling with UC. Figure 1 and Figure 2 show the most appropriate dietary recommendations during exacerbation and remission, supported by the research conducted to date, with particular emphasis on the Mediterranean diet during remission [11,12,24]. It is apparent that there is a significant difference in the diets of the two cases, so decisions about the diets of those with the disease should be made with particular care.

## 4. Unconventional Diets

### 4.1. Ulcerative Colitis Exclusion Diet (UCED)

The elimination diet specifically formulated for patients with UC may also be a future, but not yet sufficiently proven, diet. To date, its efficacy has only been evaluated in a prospective, multicenter pilot study conducted in 2021, providing a basis for further research [52]. UCED is a dietary approach aimed at modifying the composition of the gut microbiota. This diet excludes the consumption of products that may adversely affect goblet cells, the intestinal mucosa, and the composition of the gastrointestinal microbiota [52]. UCED reduces total protein intake, particularly aimed at minimizing exposure to sulfur-containing amino acids, limits intake of animal fats, saturated fatty acids, and polyunsaturated fatty acids, as well as heme and food additives. Instead, it focuses on increased intake of monounsaturated fatty acids, tryptophan, pectin, and resistant starch from natural sources. UCED is therefore a low-protein, low-fat, high-fiber, and additive-eliminating diet [52]. In the first phase of the diet, which lasts 6 weeks, the products can be divided into those allowed to be eaten in unlimited and in limited amounts, as well as those contraindicated, as shown in Table 1. However, in the second phase, which lasts from week 7 to week 12, the diet gives the patient more freedom of choice. Selected legumes are introduced, a greater variety of vegetables can be consumed, and the intake of cereal products increases [52].

A study on UCED by Sarbagili-Shabat et al. included 24 pediatric patients diagnosed with exacerbation of UC (Paediatric Ulcerative Colitis Activity Index—PUCAI > 10) of mild or moderate form [52]. Patients followed the UCED diet for 6 weeks, then, if remission occurred, continued the diet for another 6 weeks. If patients did not improve by week 3 of the diet, or if they improved but relapsed between weeks 6 and 12, a 14-day course of antibiotics (amoxicillin, metronidazole, and doxycycline) was introduced. After the pharmacotherapy period, subjects were followed up for another 7 days [52]. 

Patients received training on the diet prior to the study and were given the necessary recipes for meal preparation and recommended menus. Patients were evaluated before dietary changes were made and again at weeks 3, 6, and 12. The efficacy of the diet was tested by assessing the PUCAI index, which reached values above 10 before dietary changes were implemented, whereas disease remission was confirmed when PUCAI < 10 [52]. Nutrient supply was assessed before the dietary changes and after the study period, and it was observed that there was a reduction in total protein intake, including sulfur-containing amino acids, iron, and saturated fatty acids, in favor of monounsaturated fatty acids and dietary fiber, according to the diet. Quantitative changes in the supply of the listed components are shown in Table 2.

After 6 weeks of UCED, clinical remission was observed in nearly 37% of subjects. Some patients required additional antibiotic therapy and after 3 weeks of combined pharmacotherapy and diet, 50% achieved clinical remission. The average PUCAI value was reduced from 35 to 12.5 (pre-intervention value and at week 6, respectively). In contrast, median fecal calprotectin decreased from 818 μg/g to 592 μg/g, respectively. Thus, it is suggested that the UCED may be an effective way to induce remission in pediatric patients with mild to moderate UC [52]. While this study is promising, it is worth noting that larger studies should be conducted, particularly in adults. The first stage of the diet is very strict and limits the consumption of many foods, so it should be analyzed whether it negatively affects the nutritional status of the patient. In addition, it is a diet in which the intake of dietary fiber is increased, which is not always beneficial for patients with UC, especially during exacerbation, so it should be considered to analyze the intake of this component, especially in terms of its fraction provided with the diet. After careful consideration of all issues of dietary introduction, the group in whom it may be used, and the impact on patients’ clinical status, the UCED diet may be a suitable therapeutic tool in patients with UC. 

### 4.2. Specific Carbohydrate Diet (SCD)

The SCD is one of the most common dietary approaches in IBD reported in the literature. However, the amount of evidence for the effectiveness of the SCD diet in controlling inflammation is still small [53]. This diet is based on the premise that polysaccharides and disaccharides, through low absorption in the gastrointestinal tract, cause imbalance of the intestinal microbiota and damage to the gut, which is the cause of celiac disease and IBD [54]. As a result of this thesis, processed and canned foods, milk, and most grains, including rice, corn, wheat, and barley, are excluded from the menu [54]. The main components of the diet are simple sugars, for example, fructose, glucose, and galactose. These carbohydrates are readily absorbed, thus counteracting excessive microbial growth in the gut and dysbiosis [53,54]. Fresh fruits and vegetables, nuts, yogurt, meat, and hard cheeses are some of the recommended sources of nutrients in the SCD diet [54]. The SCD diet was first used in the first half of the 20th century by Dr. Sidney Has to treat celiac disease in children [53]. The effect of dietary specific carbohydrates on inflammation in IBD was tested in a prospective case–control study by Suskind et al. [55]. The study included 12 children aged 10–17 years with mild to moderate IBD activity. All subjects included in the analysis were asked to follow the SCD for 12 weeks. Patients’ laboratory tests were analyzed before dietary changes and at weeks 2, 4, 8, and 12. After 12 weeks, the researchers noted that recruited patients had improved disease activity index (in UC patients, the mean PUCAI value decreased from 28.3 to 6.7), decreased C-reactive protein levels, and normalized serum albumin levels. In conclusion, the authors of this study suggest that this diet may have a positive effect on clinical assessment and laboratory findings in IBD patients and is likely associated with improved gut microbiota composition [55]. The beneficial effect of SCD on gut microbiota, in individuals diagnosed with IBD in remission, was also found in an analysis by Kakodkar et al. [56]. This study additionally noted that patients with IBD in remission following a diet of specific carbohydrates reported a reduction in symptom sensation, and the ability to suspend pharmacological treatment [56]. A prospective study in pediatric patients with Crohn’s disease was also conducted to examine the significance of SCD on clinical presentation and intestinal mucosal status [57]. Sixteen patients were included in the study and were followed up for 52 weeks. Patients’ clinical status, degree of disease activity (via the Pediatric Crohn’s Disease Activity Index—PCDAI, Harvey–Bradshaw scale, and Lewis scale), and mucosal status using capsule endoscopy were assessed before follow-up and at weeks 12 and 52. The results showed that the SCD has a great positive effect on the clinical picture and intestinal mucosal status in pediatric patients with CD [57]. Another large, survey-based study by Suskind et al. involving 417 adult and pediatric patients with IBD (43% were diagnosed with UC) found a positive effect of the SCD on the health status of the subjects [58]. Researchers observed a reduction in the frequency and intensity of disease symptoms, particularly in abdominal pain and the occurrence of diarrhea, improved laboratory results in 47% of respondents, and clinical remission was observed in 33% of subjects after 2 months of the SCD, with 42% of subjects experiencing remission after 6 and also after 12 months of the elimination diet [58]. It is noted that there are few studies on the effect of dietary specific carbohydrates on IBD, and most of the existing studies are specifically on children and Crohn’s disease [54]. Additionally, the application of the results of these studies is limited because they are either case studies or retrospective in nature. Studies have shown that the SCD may have a positive effect on IBD; however, larger prospective studies, particularly in adults, are needed to recognize the efficacy and safety of this diet [54,55,56].

### 4.3. Low FODMAP

FODMAPs are fermentable oligosaccharides, disaccharides, monosaccharides, and polyols that are not absorbed in the gastrointestinal tract and thus contribute to gastrointestinal symptoms [59]. Commonly, these compounds are found in fruits, honey (which contain fructose), dairy products (abundant in lactose), onions, garlic, and wheat (containing fructans), some grains, seeds, nuts, and legumes (rich in oligosaccharides), as well as fruits and vegetables and sugar-free products that are rich in polyols such as sorbitol, xylitol, and mannitol [60,61]. The use of this diet should always be under the close supervision of a dietician, as improper use can contribute to serious nutritional disorders [62,63]. The low FODMAP diet is divided into three phases. In the first phase, FODMAPs below the threshold value are eliminated, then a challenge trial is performed, i.e., the products eliminated in the first phase are included one by one, so as to determine the impact of the different types of FODMAPs and their amounts. Finally, the development of a long-term, individually tailored diet for the patient is carried out [63]. The low FODMAP diet has found application in the treatment of *Irritable Bowel Syndrome* (IBS) [62]. After incorporating the diet, patients noticed a reduction in symptoms such as abdominal pain, bloating, diarrhea, and constipation. IBD is often accompanied by symptoms typical of IBS, so the possibility of using the diet in this group of patients has been suggested [62]. 

The positive effect of a low FODMAP diet in relieving gastrointestinal symptoms associated with IBS was demonstrated in a prospective, randomized study conducted by Pedersen et al. [64]. In the study, 89 patients (including patients in remission or with mild to moderate exacerbations of IBD) were randomly assigned to either a FODMAP-restricted diet or a standard diet for 6 weeks. Finally, the results of 78 individuals recruited for the study were analyzed. It was noted that in those following an elimination diet, up to 81% of individuals achieved dietary change effects, compared with the normal diet group where a response was seen in 46% of individuals. After the intervention period, a definite decrease in the index assessing the occurrence of IBS symptoms was noted in those following the low FODMAP diet, compared with the other group. However, this was for patients in remission and not those with mild to moderate IBD exacerbation. The main improvements were a decrease in the time and severity of abdominal pain, a decrease in bowel frequency, and an improvement in stool consistency. Additionally, a greater improvement in the quality of life of IBD patients on a low FODMAP diet has been reported [64]. The benefit of including a low FODMAP diet in the course of IBD was also found in three retrospective case-control studies [65,66,67]. In a study conducted in Australia, in IBD patients following a low FODMAP diet, an improvement in symptoms such as abdominal pain and bloating, diarrhea, and gas was found in about one out of two patients analyzed [65]. Another retrospective study conducted in 49 patients suffering from IBD revealed that about 40% of individuals signal the full effectiveness of a long-term low FODMAP diet. The main complaints reported to improve were abdominal pain and bloating [66]. A recent review of the medical records of 88 people with IBD by Prince et al. showed that the inclusion of a low FODMAP diet not only reduces the severity of gastrointestinal complaints, but also has a positive effect on stool consistency and frequency of bowel movements [67]. These results show that a low FODMAP diet may be associated with relief of gastrointestinal complaints in patients with IBD, although it does not affect inflammation in the gut [54]. However, it is very important that elimination of products does not last for too long a period of time, as this can lead to nutrient deficiencies and malnutrition [53]. The diet is also unfavorable if applied long-term due to possible negative effects on the gut microbiota of individuals with IBD [53]. 

FODMAPs are prebiotics, which are nutrients for gastrointestinal microbes [68,69]. It is suspected that by eliminating products in the diet that provide these compounds, one is exposed to reduced food for the gut microbiota. This may reduce the production of short-chain fatty acids, which lower the pH in the lumen of the gastrointestinal tract, modulate the functioning of the immune system, and thus may have a positive effect on human health [69]. Two studies by Halmos et al. evaluated how diets with different FODMAP contents may affect the microorganisms present in the human gastrointestinal tract [68,70]. One of these randomized controlled trials compared the effects of a FODMAP-restricted diet and a standard “Australian” diet on colonic microbiota composition and biomarkers related to intestinal mucosal function, in subjects with established IBS (27 subjects) and a population of healthy subjects (6 subjects) [68]. Before the intervention, subjects were asked to record their typical diet for 7 days and a stool sample was collected from them for 5 days. Subjects were then randomly assigned to one of the groups, the first to follow a low FODMAP diet (average 3.05 g/d) for 21 days, and the second to follow a standard “Australian” diet containing FODMAPs (at an average of 23.7 g/d) for the same period of time. Throughout the intervention period, fecal collections were made, and comparative analysis of the collected materials was performed. The concentration of short-chain fatty acids, pH value, and the number and diversity of colonic microorganisms were checked. Higher pH values and lower bacterial counts were noted in subjects limiting FODMAP intake compared with those on a typical diet. Additionally, butyric acid bacteria and bacteria associated with normal mucosal structure were found to be more abundant in those on the standard diet than those in the low FODMAP group. Thus, from the results, it can be concluded that the introduction of restrictions on FODMAP consumption should be carried out carefully, with an evaluation of the pros and cons factors, because it may have a negative impact on the presence of microorganisms in the gastrointestinal tract that are beneficial to human health [68]. Another study led by Halmos et al. was conducted in patients with Crohn’s disease in remission [70]. As in the previous study, patients were randomly assigned to either a low FODMAP diet or a typical “Australian” diet for 21 days. For the last five days of each intervention, stool samples were collected from each subject and evaluated for pH values, fecal calprotectin and short-chain fatty acid concentrations, and the number of microorganisms colonizing the gut. In addition, the gastrointestinal symptoms present were recorded daily. Although there was no difference in pH, short-chain fatty acid concentration, or total microbial count between subjects on the low FODMAP diet and those on the standard “Australian” diet, a difference in microbiota diversity was noted. The FODMAP-restricting subjects had fewer butyric acid-producing bacteria and affected normal mucosal structure, compared with the other group. Differences in diet were not significant in fecal calprotectin concentrations, but gastrointestinal complaints were increased in those on the “Australian” diet. Thus, the results of this study show that the use of a low FODMAP diet for a prolonged period of time may contribute to dysbiosis, shows no effect on intestinal inflammation, and despite its potential positive effects on the symptoms of IBD, its use in this disease entity is not fully justified [70]. A review of studies by Vandeputte et al. also found that limiting FODMAP intake in individuals with intestinal disease may contribute to reduced *Bifidobacterium* and other changes in the gut microbiota comparable to dysbiosis [69]. However, there is no clear answer as to whether these changes are irreversible or only occur during the period of severe dietary restriction occurring during the diet [69]. In conclusion, despite the promising results of the low FODMAP diet in alleviating gastrointestinal symptoms, this diet needs more research to ensure its safety in patients with IBD.

Establishing accurate dietary recommendations for patients with IBD is invariably a great challenge because there is insufficient research to suggest the unequivocal use of any currently popular diet. It is certain that there is a need for further, particularly randomized, clinical trials that could help to better understand the mechanisms of dietary effects on IBD.

## 5. Supplementation

### 5.1. Iron

Iron is an essential element for every cell in the human body and thus necessary for the proper functioning of all organs [71]. Iron can have a particularly strong effect on the health of people with chronic inflammatory diseases. Its deficiencies are likely to contribute to the severity of the condition and, with it, cause an accelerated exacerbation of the patient’s clinical condition [71]. 

Iron deficiency occurs on average in 13%–90% of patients with IBD [71]. Iron deficiency is one of the most commonly identified parenteral complications of the disease and one of the most common causes of anemia in patients with IBD [12,71]. It is widely accepted that even a slight iron deficiency is classified as anemia, whereas actually anemia is the result of extreme iron deficiency [71]. Symptoms of iron deficiency are often present in patients prior to the onset of anemia, but for the most part, they are overlooked by patients and their physicians, and thus corrective measures are not taken [71,72]. The signs of iron deficiency are uncharacteristic, including chronic fatigue and weakness [71]. 

A significant influence on the occurrence of iron deficiency in patients with IBD is the reduced iron absorption caused by ongoing chronic inflammation in the gastrointestinal tract or intestinal resection, the patient’s malnourished state, and blood loss [73,74]. 

It is suggested that all individuals with IBD should be screened for anemia, regardless of age [12,73]. For diagnostic purposes, readily available blood tests are used, which include blood count and serum ferritin and C-reactive protein levels, as well as transferrin saturation (TSAT) [12,71]. This follow-up is performed every 6–12 months in case of disease remission or mild UC. In contrast, patients with active inflammation should be measured at a maximum of every 3 months [73]. Ferritin levels <30 μg/L are the basis for the diagnosis of iron deficiency anemia in patients with UC without clinical, biochemical, or endoscopic signs of active inflammation. Ferritin levels below 100 μg/L or TSAT <20% may reflect iron deficiency in patients with active IBD [12,71]. 

Although there are emerging studies suggesting positive effects of iron supplementation, in iron deficiency states without the presence of anemia, in some conditions, such as chronic fatigue and heart failure, there are currently no data relating to IBD [12]. Therefore, it is considered unclear to administer iron preparations in the absence of anemia; additionally, this will be determined by the patient’s medical history and presenting symptoms [12]. If iron deficiency anemia is found, iron supplementation should be instituted immediately [12,73]. The 2015 guidelines set by the European Crohn’s Disease and Colitis Organization (ECCO) note that faster and more effective patient outcomes may occur after intravenous iron supplementation than after oral administration, and that this form of iron deficiency compensation is better tolerated by the patient [75]. Correcting iron deficiency in patients with UC has a significant impact on improving the quality of life of these individuals [71,74].

### 5.2. Vitamin D and Calcium

Vitamin D has been suggested to have a significant effect on immune function, mainly on non-specific immunity, and its potential impact on the development of IBD and its use in the treatment of IBD have been highlighted [76].

Laboratory testing is necessary to determine insufficient vitamin D levels in the human body, as vitamin D deficiencies usually do not produce clear symptoms [76]. The best exponent for determining the amount of vitamin D is the determination of serum 25 hydroxyvitamin D [25(OH)D] levels [76]. Normal saturation of the body with vitamin D is found at a concentration of 30–100 ng/mL 25(OH)D, while lower values indicate deficiency [76]. 

Vitamin D deficiency is found in 10–75% of patients with IBD [77]. Insufficient levels of this vitamin can be found twice as often in people with UC as in healthy individuals [78].

People with IBD have an increased risk of fractures by up to 40% compared with healthy people [79]. The risk of osteoporosis is high in patients with IBD and is particularly increased in patients treated with corticosteroids [76,79]. Additionally, inadequate vitamin D levels in the human body may correlate with increased intestinal membrane permeability and impaired immune function, which may not only contribute to the development of IBD but also increase the risk of relapse [78].

Causes of vitamin D deficiency may include low sun exposure, inadequate intake of vitamin D source products, impaired absorption by inflammation occurring in the gastrointestinal tract or intestinal resection, increased absorption by cells that are inflamed, and impaired renal metabolism and increased catabolism and excretion [76,77,78].

Although ECCO indicates that there are insufficient data to mandate routine vitamin supplementation, due to the possible association of low human vitamin D concentrations and possibly worse outcomes in the course of IBD, the use of supplementation in these patients may be important [77,80]. A small, prospective, randomized study by Mathur et al. showed that vitamin D supplementation in UC patients with vitamin D deficiency can improve patients’ quality of life and minimize inflammatory activity, especially at doses of 4000 IU/day compared with 2000 IU/day [81]. An analysis by Gubatan et al. showed that serum vitamin D levels <35 ng/mL during remission likely increase the risk of relapse in patients with UC [82]. It has also been noted that the lower the 25(OH)D levels in patients with UC, the greater the mucosal inflammation [83]. Individuals with low 25(OH)D levels had greater disease severity and higher prevalence [84]. Additionally, vitamin D supplementation is indicated because of the effect of low blood levels of vitamin D on the increased risk of surgery for UC. Low vitamin D levels are important in increasing the risk of *Clostridium difficile* infection [12]. However, there is no exact consensus on the appropriate vitamin D levels in UC patients [12,80].

The use of vitamin D supplementation is safe and well tolerated by patients; therefore, although the evidence regarding the involvement of vitamin D in the course of UC is not strong enough, supplementation should be used in all patients with IBD in case of deficiencies of this vitamin [85]. Guidelines from ESPEN, as well as the Endocrine Society, indicate that people with IBD should always be screened for serum vitamin D and calcium levels, and adequate supply of these components should be ensured. This is extremely important in patients treated with corticosteroids [12,86]. It is recommended that vitamin D levels should be reassessed once a year or once every two years in patients with active inflammation in the course of UC, confirmed metabolic bone disease, or chronic use of corticosteroid therapy [76]. To achieve serum 25(OH)D levels >30 ng/mL in patients with known vitamin D deficiency, treatment is based on the administration of vitamin D_3_ or D_2_ for eight weeks of 6000 IU/day or 50,000 IU 1×/week for adults [86].

### 5.3. Vitamin B_9_

Vitamin B_9_ deficiency in individuals with IBD may be caused by inadequate intake with the diet, impaired absorption by inflammation occurring within the gastrointestinal tract, increased utilization by inflamed cells, and drug interactions, particularly methotrexate or sulfalazine [12,87]. Sulfalazine affects vitamin B_9_ deficiency by causing malabsorption syndrome. In contrast, methotrexate reduces the action of dihydrofolate reductase, which is required for the conversion of dihydrofolic acid to tetrahydrofolic acid [12].

A meta-analysis by Pan et al. showed that folic acid deficits are significantly more common in UC patients compared with the healthy population [87]. A retrospective study conducted in Spain also presents that people with IBD face vitamin B_9_ deficiency quite often [88]. 

A dangerous complication of IBD is an increased risk of colorectal cancer. A meta-analysis of 10 studies by Burr et al. demonstrated that folic acid supplementation may have a defensive effect against the development of colorectal cancer in patients with IBD [89].

ESPEN guidelines indicate that any patient with active inflammation who is being treated with sulfalazine or who has macrocytosis should be routinely monitored for serum folic acid and red blood cell levels. These recommendations also suggest that some patients with IBD, including those treated with sulfalazine or methotrexate, should receive folic acid supplementation [12,73].

### 5.4. Omega-3 Fatty Acids

Supplementation of omega-3 fatty acids can be used in the treatment of some autoimmune diseases, including rheumatoid arthritis [90]. In these diseases, there is an imbalance between the production of free radicals and the levels of antioxidants that can reduce tissue damage and inflammation [77]. Supplementation with omega-3 fatty acids, which have shown potential anti-inflammatory effects, may be beneficial in this situation. These components reduce the production of IL-1, IL-6, and tumor necrosis factor-alpha (TNF-alpha) [77]. However, for some inflammatory diseases, such as in IBD, the utilization of the properties of these fatty acids and their use as supplementation is uncertain [35,90]. 

One study showed that supplementation with omega-3 fatty acids, in patients in remission from UC, reduced the rate of relapse at months 2 and 3 of the study; however, at the end of the study, the number of relapses, compared to the control group, was not reduced. The authors of this study suggest that these acids do not prevent recurrence but only delay its onset [91]. Two systematic reviews have shown that omega-3 fatty acids have no positive effect in supporting remission maintenance in UC [79,92]. Therefore, ESPEN guidelines do not recommend supplementation of these compounds in patients with IBD to maintain remission [12].

### 5.5. Probiotics

The term probiotics includes live microorganisms that, when taken in optimal amounts, have positive health effects in the host [93]. Reducing the growth of pathogenic microorganisms, improving intestinal barrier function, and modulating the body’s immune system response are the main proposed mechanisms of action of probiotics [94].

The probable beneficial effects of probiotics on human health can be exploited in gastrointestinal diseases, including IBD [94]. A meta-analysis of randomized controlled trials by Fujiya et al. indicated that administration of probiotics may be beneficial in the treatment of UC during the remission period of the disease as well as during the acute phase of UC [94]. Probiotics may prevent recurrence of acute flare-ups, and during UC flare-ups they help to soothe inflammation. Their action, when used to maintain remission, is comparable to that of mesalazine [94]. Another meta-analysis evaluating the effect of probiotic therapy on the course of UC was conducted by Shen et al. [95]. The researchers analyzed a total of 23 randomized control trials with a total of 1763 patients with UC. The results showed that the use of probiotics in patients with exacerbation of UC significantly increases the number of clinical remissions. However, the only effective preparation was VSL#3. It is also worth noting that this probiotic in clinical remission significantly reduces the recurrence of active disease [95]. A recent meta-analysis of randomized controlled trials evaluated the results of 14 studies (12 conducted in adults, 2 in pediatric patients) involving a total of 865 individuals with mild to moderate UC [96]. The studies reviewed confronted the use of probiotics with placebo or with mesalazine, as well as probiotics combined with masalazine and the use of masalazine alone. Half of the studies used a single strain, while the others used mixtures of probiotic strains. Observations lasted from 2 to 52 weeks. The results showed that in combination with placebo, probiotics can induce clinical remission. The clinical effects of probiotics and mesalazine were similar. In combination therapy with mesalazine, probiotics can have a small positive effect on inducing clinical remission. However, the authors of the meta-analysis indicated that the scientific evidence obtained so far has low reliability and is limited to selected cases. Therefore, it is believed that there is a need for carefully designed randomized control trials. A special attempt should be taken to increase the power of the analyses performed and to standardize the selection of study groups [96]. 

A randomized clinical trial conducted by Miele et al. demonstrated that the use of VSL#3 in children with UC may be effective and safe to induce remission and may subsequently assist in the maintenance process [97]. Another prospective, randomized clinical trial in 2012 presented that the use of rectal enemas containing *Lactobacillus reuteri* ATCC 55730, during the active phase of mild to moderate end-stage UC in children, has positive outcomes. These infusions helped to calm inflammation in the colon and modify mucosal secretion of cytokines that are involved in IBD [98].

On the basis of the study findings, ESPEN recommendations advise against the use of probiotics in the therapy of active stage of severe UC due to the possibility of bacteremia and lack of data on their effectiveness [12]. These guidelines, however, do not eliminate their use when there is a desire to induce remission in mild to moderate UC [12]. At the same time, ESPEN points out that not all probiotics can be used. There is the possibility of including specific preparations, including *Escherichia coli* Nissle 1917 or VSL# 3 [12].

There are no clear recommendations for vitamin and mineral supplementation for patients with UC. However, patients with UC, due to the inflammation present in the intestinal mucosa, are more likely than healthy individuals to be deficient. For this reason, it is undeniable that every patient should undergo routine biochemical testing to determine any deficiencies that may exist, in order to incorporate appropriate supplementation as soon as possible and restore normal values of biochemical indices. In particular, attention should be paid to vitamins and minerals that are most commonly found to be deficient, such as iron, vitamin D, and folic acid. Due to the lack of sufficient clinical trials, patients are not recommended to take supplementation of specific ingredients, including omega-3 fatty acids and probiotics. However, these compounds, due to their possible positive effects on the course of the disease, should be more extensively evaluated in terms of their use in patients with UC, as patients could thus particularly benefit in improving their quality of life and test results.

## 6. Conclusions

The Mediterranean diet has the greatest therapeutic potential through the selection of foods containing ingredients that influence the gut microbiota, colonocytes, and body function. It is reasonable, after taking into account individual modification, to recommend this diet to patients with UC. The diet is worth reviewing more broadly in terms of its benefits for people with UC in both remission and exacerbation.Unconventional diets, such as low FODMAP and the SCD, should not be recommended to patients with UC because their use may lead to a disruption of the patient’s nutritional status, while evidence of beneficial effects is still scarce.The UCED diet, when appropriately tailored to the patient and with accurate recommendations, may prove to be a powerful therapeutic tool in people with UC. Therefore, this diet is a good basis for more research, with both children and adults included.Regular monitoring of vitamin and mineral serum concentrations is a very important element of treatment of patients with UC, and in case of deficiencies, appropriate supplementation should be instituted. Studies conducted so far show promising results with omega-3 fatty acid supplementation and probiotics, but further studies need to be conducted to conclusively confirm their effectiveness.

## Figures and Tables

**Figure 1 nutrients-14-02469-f001:**
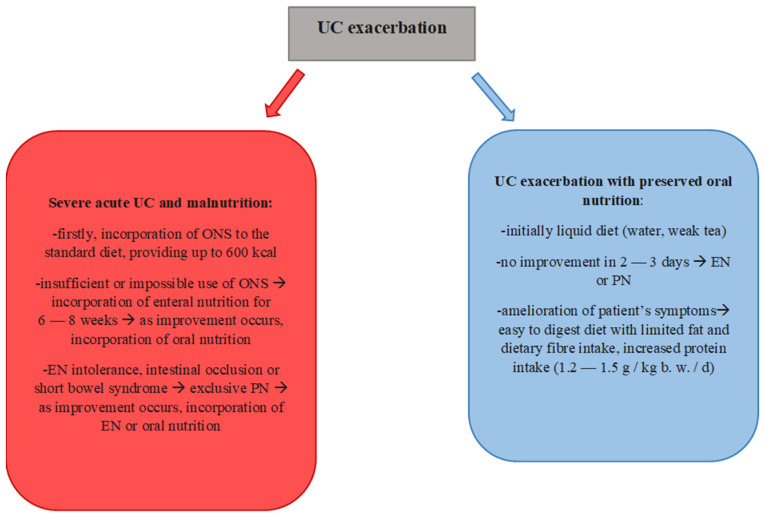
General dietary recommendations for patients during UC exacerbation [11,12]. (**ONS**—Oral Nutritional Supplements, **EN**—Enteral nutrition, **PN**—Parenteral nutrition).

**Figure 2 nutrients-14-02469-f002:**
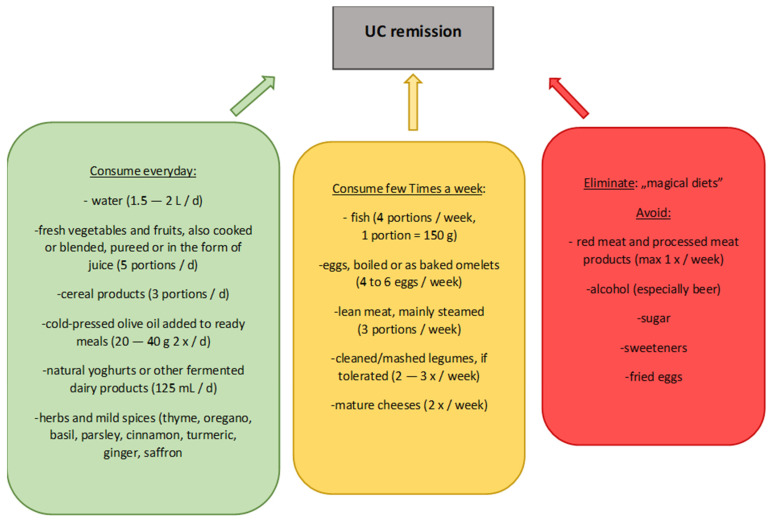
General dietary recommendations for patients in UC remission.

**Table 1 nutrients-14-02469-t001:** Recommended intake of foods in phase 1 of UCED [52].

Allowed in Unlimited Quantities	Allowed in Limited Quantities	Contraindicated
Vegetables	Eggs	Red meat
Fruits	Poultry	Processed foods
Rice	Yoghurts	Sugar
Potatoes	Pasta	

**Table 2 nutrients-14-02469-t002:** Changing the amount of nutrients consumed [52].

Nutrients	Ingestion Prior to UCED Application	Ingestion after 6 Weeks of UCED
Total protein	1.8 g/kg b.w./d	1.2 g/kg b.w./d
Cysteine	0.8 g/d	0.5 g/d
Methionine	1.6 g/d	0.9 g/d
Iron	12.1 mg/d	8.7 mg/d
Saturated fatty acids	19.5 g/d	8.3 g/d
Monounsaturated fatty acids	21.6 g/d	27.3 g/d
Dietary fiber	16.4 g/d	21.7 g/d

## Data Availability

Not applicable.

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
