# Peer review of "Nutrition and Supplementation in Ulcerative Colitis"

_nutrients, 2022, doi:10.3390/nu14122469_

Round 1

Reviewer 1 Report

In the current Review Article titled as "Nutrition and supplementation in ulcerative colitis" the authors aimed to analyse and summarize the current knowledge on the impact of various nutritional plans and dietary supplements in ulcerative colitis.

The manuscript is well and thorough written. 

Reviewer 2 Report

Review

Nutrition and supplementation in ulcerative colitis

In the abstract:

-          Ulcerative colitis should not be italic in the abstract or text.

-          Author should mention one sentence about the strategy for searching the literatures.

-          Authors should rewrite the abstract and follow the normal flow meaning that the aim of the study should mention at the beginning of the abstract and not at the end (introduction, aims, methodology, findings and conclusion).

-          After the introduction, author should add one section named methods and write in details about the strategy followed for collecting the suitable articles, mentioning the years and the key words used for collecting these suitable articles.

-          Introduction is well written and designed.

-          The study answered the research question.

Important comments:

-          Conclusion should be rewritten properly.

-          References should be written according to the journal guidelines as most of them are not according to the journal instructions.

Reviewer 3 Report

This is a review by Radziszewska et al, the authors provide a thorough and well-balanced review of diet and nutritional supplementation in patients with inflammatory bowel disease (IBD), with more emphasis on colitis (ulcerative colitis, or UC). Also, while other specialized diets are addressed in the review, discussion on the benefits of the Mediterranean diet (MD) in the UC population is given more focus, which the authors aptly provided justification for. Given this emphasis, a search of pubmed was conducted on MD and colitis, with results showing 36 research-based publications and 11 reviews. The topic is timely and of great importance therefore, and the authors provide an updated list of studies/references to consider (within the last 1-5 years) which help justify the need for publication.

Overall, the review is extremely well-written and the organization is quite strong. Only a few minor suggestions were noted, which include the following:

1. Consider combining the first 3 paragraphs of the Introduction section (lines 25-32).

2. Lines 98-103: There is a little bit of confusion and possible contradiction in the following sentences about the microbiome, particularly Bacteroides. In lines 98-99 Bacteroides are considered "beneficial", whereas in line 102, the genus is considered "unfavored". Of course, this is accurate as different species within this genus can be favorable or unfavorable. However, in the current review it is confusing to have these contradictory statements back-to-back. A suggestion would be to mention some Bacteroides have been noted to be harmful or induce colitis disease severity (Bacteroides vulgatus and Bacteroides thetaiotaomicron, see PMID:21575910 ), while others have been know to display protective effects (Bacteroides fragilis, see PMID: 34308455).

3. It might be better to combine paragraph starting at Line 289 with the next paragraph (Line 292).

4. Please define abbreviations ONS (oral nutritional supplements?) EN and PN from Figure 1 if not present in the main manuscript.

Author Response

Please see the attamchment.
